rsos.royalsocietypublishing.org

organic chemistry/green chemistry/physical chemistry

dicationic ionic liquids, homogeneous, heterogeneous, miscibility

**Authors for correspondence:**
Bi-Xian Zhang
e-mail: zg_2008@163.com
Xiao-Mei Hu
e-mail: huxiaomei1982@163.com

This article has been edited by the Royal Society of Chemistry, including the commissioning, peer review process and editorial aspects up to the point of acceptance.

# Synthesis and investigation of physico-chemical properties of dicationic ionic liquids

Yi-Xin Sun[1], Ying-Ying Wang[1], Bing-Bing Shen[1], Bi-Xian Zhang[2] and Xiao-Mei Hu[1]

[1]College of Life Science, Northeast Agricultural University, Harbin 150030, People's Republic of China
[2]Heilongjiang Academy of Agricultural Sciences, Harbin 150086, People's Republic of China

X-MH, 0000-0001-7191-0915

A series of dicationic ionic liquids (ILs) including $[C_4(MIM)_2][PF_6]_2$, $[C_5(MIM)_2][PF_6]_2$, $[C_6(MIM)_2][PF_6]_2$ and $[C_4(PYR)_2][PF_6]_2$ were synthesized. Their thermal stability and melting points were analysed. It was found that dicationic ILs presented important implications in the design of homogeneous and heterogeneous system with water. A homogeneous system of dicationic ILs with water could be formed at a relatively high temperature and then a heterogeneous system was formed when the solution was cooled to a low temperature. The ILs recovered by altering the temperature were obtained in high percentage yields of $[C_4(MIM)_2][PF_6]_2$ (97.6%), $[C_5(MIM)_2][PF_6]_2$ (97.3%), $[C_6(MIM)_2][PF_6]_2$ (98.0%) and $[C_4(PYR)_2][PF_6]_2$ (94.2%). On the other hand, $[C_4(MIM)_2][PF_6]_2$ and $[C_5(MIM)_2][PF_6]_2$ exhibited good solubility in acetonitrile and acetone. A homogeneous system could be achieved with imidazolium-based ILs with a relatively low amount of water and acetonitrile at room temperature. All of the properties of dicationic ILs have a strong correlation with the nature of dications, the linkage chain and the symmetry of dications. Dicationic ILs may provide a new opportunity for some specific applications in order to enable the effective separation and isolation of products.

## 1. Introduction

Ionic liquids (ILs) have received wide popularity in recent years in the areas of electrochemistry [1], catalytic reactions [2], organic synthesis [3], separation technology [4] and biochemistry [5]. ILs attract research interests because they possess a number of unique properties such as high thermal stability [6], no flammability [7], low vapour pressure and ease of reuse [8,9]. Moreover, the physical and chemical properties can be adjusted by altering their chemical structure of cations and anions [10,11].

rsos.royalsocietypublishing.org    R. Soc. open sci. **5**: 181230

Commonly, ILs are formed by an organic cation such as imidazolium, pyridinium, ammonium and phosphonium with an inorganic anion such as $Cl^-$, $Br^-$, $PF_6^-$, $BF_4^-$ and $NTf_2^-$ (bis (trifluoromethanesulfonyl) amide). ILs are classified into water-miscible ILs (hydrophilic) and water-immiscible ILs (hydrophobic) [12]. The miscibility of ILs with water depends on their anions [13]. ILs with the anion of $PF_6^-$ and $NTf_2^-$ are water-immiscible, whereas others are completely water-miscible. However, an increase in the alkyl chain length of ILs cation could increase their hydrophobic character.

Recently, dicationic ILs with two monocations combined into one cation have been synthesized. Owing to the strong interaction between the cation and the anion in the dicationic compound, many dicationic salts generate melting points higher than 100°C [14]. Several ammonium-based dicationic ILs were synthesized by Engel *et al.* [15–17]. Some imidazolium-based dicationic ILs were prepared by Ohno *et al.* [18]. Anderson and co-workers reported 39 types of dicationic ILs [14].

Dicationic ILs generate good thermal stability [19], high conductivity and low electrochemical window [20]. They have been shown to possess superior physico-chemical properties as solvents of high temperature organic synthesis or as a gas chromatography stationary phase, because these applications were often conducted at relatively high temperatures [21–24]. Studies have shown that their physical and chemical properties could be affected significantly by controlling the length of the alkyl side chains connecting the two cations [14]. Dicationic ILs were employed for many applications including dye-sensitized cells [25], organic synthesis [26], lubricants and lubricant additives [27].

In current studies, ILs are reported either water-miscible or water-immiscible. ILs based on the anion of $PF_6^-$ and $NTf_2^-$ are water-immiscible. There are no reports concerning the variable miscibility between ILs and water. In this study, dicationic ILs comprising imidazolium-based and pyridinium-based dications were synthesized. The main goal of this investigation was to increase knowledge in physico-chemical properties of dicationic ILs. The main attention was focused on the investigations of (a) its thermal stability, (b) its melting points and (c) its solubility in water and water-organic mixtures. The properties of dicationic ILs including thermal stability, miscibility with water and water-organic mixtures could be important for their use as solvents or catalysts in various applications.

# 2. Experimental

## 2.1. General

All chemicals were provided by Aladdin Company (China) and were used as received: 1-methyl imidazole (99%), pyridine (99.5%), 1,4-dichlorobutane (99%), 1,5-dichloropentane (99%), 1,6-dicholorohexane (99%) and $KPF_6$ (99%). $^1H$-NMR was recorded on an AV-III 400 MHz spectrometer (Bruker, Germany). ILs were dried in a DZF-6050 vacuum (Shanghaiyiheng, China).

## 2.2. Synthesis of dicationic ILs

### 2.2.1. [1,1′-(butane-1,4-diyl)-bis(3-methylimidazolium)] di(hexafluorophosphate) ([C$_4$(MIM)$_2$][PF$_6$]$_2$)

1,4-dichlorobutane (11.17 ml, 0.1 mol) was added dropwise into 1-methylimidazole (15.95 ml, 0.2 mol) in acetonitrile (30 ml). The mixture was stirred at 80°C for 36 h. The resulting solid was filtered and washed by acetonitrile (3 × 20 ml). The trace of acetonitrile was removed by rotary evaporation at 60°C for 30 min. The solid was further dried under vacuum at 60°C for 24 h. After that, an aqueous solution of the solid (14.56 g, 0.05 mol) was dropped slowly into an aqueous solution of $KPF_6$ (20.45 g, 0.11 mol). The reaction proceeded for 24 h at room temperature. The resulting precipitates were washed with deionized water and were dried under vacuum at 60°C for 24 h to give [C$_4$(MIM)$_2$][PF$_6$]$_2$. Yield: (18.79 g, 74%). $^1H$-NMR (400 MHz, DMSO-d$_6$) (ppm): 9.061 (s, 2H, imidazolium ring), 7.708–7.729 (d, 4H, imidazolium ring), 4.187 (t, 4H, (–N–CH$_2$–)$_2$, 3.844 (t, 6H, (–CH$_3$)$_2$), 1.761 (q, 4H, –CH$_2$–CH$_2$–).

### 2.2.2. [1,1′-(pentane-1,5-diyl)-bis(3-methylimidazolium)] di(hexafluorophosphate) ([C$_5$(MIM)$_2$][PF$_6$]$_2$)

1,5-dichloropentane (12.88 ml, 0.1 mol) was added into 1-methylimidazole (15.95 ml, 0.2 mol) in acetonitrile (30 ml). The reaction was carried out at 70°C for 36 h. The resulting solid was washed with ethyl acetate (3 × 20 ml) and then was dried in a vacuum at 60°C for 24 h. Following that, an aqueous solution of the solid (15.26 g, 0.05 mol) was dropped into an aqueous solution of $KPF_6$ (20.45 g, 0.11 mol). The reaction proceeded for 24 h at room temperature. The resulting precipitates were washed with deionized water and were dried under vacuum at 60°C for 24 h to give

rsos.royalsocietypublishing.org    R. Soc. open sci. **5**: 181230

$[C_5(MIM)_2][PF_6]_2$. Yield: (15.12 g, 58%). $^1$H-NMR (400 MHz, DMSO-d$_6$) (ppm): 9.070 (s, 2H, imidazolium ring), 7.708–7.743 (d, 4H, imidazolium ring), 4.175–4.139 (t, 4H, $(-N-CH_2-)_2$, 3.854 (t, 6H, $(-CH_3)_2$), 1.798–1.836 (q, 4H, $-CH_2-CH_2-$), 1.226 (q, 2H, $-CH_2-$).

### 2.2.3. [1,1'-(hexane-1,6-diyl)-bis(3-methylimidazolium)] di(hexafluorophosphate) ($[C_6(MIM)_2][PF_6]_2$)

A similar protocol was employed as that used for $[C_4(MIM)_2][PF_6]_2$. 1,6-dichlorohexane (12.88 ml, 0.1 mol) was reacted with 1-methylimidazole (15.95 ml, 0.2 mol) in acetonitrile (30 ml). The mixture was stirred and refluxed at 80°C for 36 h. The resulting solid (15.96 g, 0.05 mol) was then reacted with KPF$_6$ (20.45 g, 0.11 mol) in an aqueous solution at room temperature for 24 h to give $[C_6(MIM)_2][PF_6]_2$. Yield: (24.33 g, 90%). $^1$H-NMR (400 MHz, DMSO-d$_6$) (ppm): 9.074 (s, 2H, imidazolium ring), 7.701–7.743 (d, 4H, imidazolium ring), 4.127–4.163 (t, 4H, $(-N-CH_2-)_2$, 3.850 (t, 6H, $(-CH_3)_2$), 1.780 (q, 4H, $-CH_2-CH_2-$), 1.226 (q, 4H, $-CH_2-CH_2-$).

### 2.2.4. [1,1'-(butane-1,4-diyl)-bis(3-methypyridinium)] di(hexafluorophosphate) ($[C_4(PYR)_2][PF_6]_2$)

Pyridine (16.22 ml, 0.2 mol) was treated with 1,4-dichlorobutane (11.17 ml, 0.1 mol). The mixture was stirred at 110°C for 36 h. The resulting solid was washed with acetonitrile and was dried under vacuum at 60°C for 24 h. After that, the solid (14.26 g, 0.05 mol) was reacted with KPF$_6$ (20.45 g, 0.11 mol) in an aqueous solution for 24 h at room temperature. The resulting product was filtered and dried to give $[C_4(PYR)_2][PF_6]_2$. Yield: (24.09 g, 96%). $^1$H-NMR (400 MHz, DMSO-d$_6$) (ppm): 9.042–9.056 (d, 4H, pyridinium ring), 8.626 (t, 2H, pyridinium ring), 8.162–8.196 (t, 4H, pyridinium ring), 4.635 (t, 4H, $(-N-CH_2-)_2$), 1.95 (q, 4H, $-CH_2-CH_2-$).

### 2.2.5. 1-butyl-3-methylimidazolium hexafluorophosphate ($[C_4MIM]PF_6$) and 1-butyl-pyridinium hexafluorophosphate ($[C_4PYR]PF_6$)

1-methylimidazole (7.97 ml, 0.1 mol) was mixed with 1-chlorobutane (20.00 ml, 0.19 mol). The mixture was stirred and refluxed at 70°C for 48 h. After that, the crude ionic liquid was washed with 1-chlorobutane (2 × 20 ml). The trace of 1-chlorobutane was removed with rotary evaporation at 60°C for 30 min, followed by at 80°C for 4 h under high vacuum to give $[BMIM]^+Cl^-$. $[BMIM]^+Cl^-$ (17.46 g, 0.1 mol) was reacted with KPF$_6$ (18.60 g, 0.1 mol) in aqueous solution. The mixture was stirred at room temperature for 24 h. The resulting $[BMIM]PF_6$ was washed with deionized water (3 × 10 ml) and kept in the high vacuum at 60°C for 3 h. Yield: (25.86 g, 91%). $^1$H-NMR (400 MHz, DMSO-d$_6$) (ppm): 9.062 (s, 1H, imidazolium ring), 7.66–7.713 (m, 2H, imidazolium ring), 4.176 (t, 2H, N–CH$_2$–), 3.857 (s, 3H, N–CH$_3$), 1.772 (q, 2H, N–CH$_2$–CH$_2$–), 1.271 (sext, 2H, N–CH$_2$CH$_2$–CH$_2$–), 0.905 (t, 3H, –CH$_3$). A similar protocol was used for $[C_4PYR]PF_6$. Yield: (15.18 g, 54%). $^1$H-NMR (400 MHz, DMSO-d$_6$) (ppm): 9.068–9.082 (d, 2H, pyridinium ring), 8.591 (s, 1H, pyridinium ring), 8.148 (t, 2H, pyridinium ring), 4.593 (t, 2H, $(-N-CH_2-)$, 1.897 (q, 2H, N–CH$_2$–CH$_2$–), 1.284 (sext, 2H, N–CH$_2$CH$_2$–CH$_2$–), 0.908 (t, 3H, –CH$_3$).

## 2.3. Thermal stability analysis

The thermal stability of dicationic ILs was determined by thermogravimetric analysis (TGA). 5–10 mg of the sample was placed in an open alumina plate by using a TG 209 F3 thermogravimetric analyser (NETZSCH, Germany). It was performed in a dynamic mode at 30°C to 600°C under a nitrogen atmosphere (50 ml min$^{-1}$). The measurement was started at room temperature with a heating rate of 10°C min$^{-1}$ [28].

## 2.4. Solubility of dicationic ILs with organic solvents

The ionic liquid (1.0 g) was mixed with one of the organic solvents (1 ml). The mixture was shaken strongly every 5 min for 30 s at room temperature (25°C). If the mixture was immiscible within 30 min, another 1 ml of the organic solvent was added into the mixture. The procedure was repeated until the mixture became a clear solution and the volumes of organic solvent required were recorded.

**Scheme 1.** Synthesis of dicationic ionic liquids.

## 2.5. Miscibility of dicationic ILs with water

The ionic liquid (1.0 g) was mixed with deionized water (1 ml). The mixture was stirred (200 r.p.m.) for 30 min at a different temperature. If the mixture was immiscible, another 1 ml of deionized water was added into the mixture, which was repeated until the mixture became a clear solution. The volume of deionized water was recorded. In addition, the ionic liquid (1.0 g) was mixed with water (1 ml) and $CH_3CN$ (1 ml). The mixture was shaken strongly every 5 min for 30 s at room temperature (25°C). If the mixture was immiscible within 30 min, another 1 ml $CH_3CN$ was added into the mixture. The procedure was repeated until the mixture became a clear solution. On the other hand, if the mixture was miscible, another 0.5 ml of deionized water was added into the solution to check whether the ILs could precipitate out. The volumes of deionized water and $CH_3CN$ were recorded.

## 2.6. The recovery of dicationic ILs

About 1.0 g of ionic liquid was mixed with a minimum amount of deionized water that was required for the complete miscibility at 100°C. After the mixture was completely miscible, the solution was cooled to room temperature (25°C). The ILs precipitated out of the aqueous solution and they were filtered out and dried in a vacuum at 60°C for 24 h to give the recovery ILs.

# 3. Results and discussion

## 3.1. Synthesis of dicationic ILs

Dicationic ILs were synthesized through commonly used quaternization reaction followed by anion metathesis (scheme 1). In the process of quaternization reaction, acetonitrile as the reaction solvent was used for the preparation of $b_1$ and $b_3$ and the resulting product precipitated out of the solution effectively. However, ethyl acetate should be employed to take the place of acetonitrile for the synthesis of $b_2$. Although $b_2$ was dissolved in acetonitrile, it could precipitate out of ethyl acetate, which was readily filtered out.

As mentioned in the Experimental section, yields of ILs followed the order of $[C_6(MIM)_2][PF_6]_2$ (90%) > $[C_4(MIM)_2][PF_6]_2$ (74%) > $[C_5(MIM)_2][PF_6]_2$ (58%). A high yield was achieved for $[C_6(MIM)_2][PF_6]_2$ and a low yield was observed for $[C_5(MIM)_2][PF_6]_2$. The possible reason could be that the anion exchange reaction was carried out in an aqueous solution. The aqueous layer was separated off and the organic layer was washed with deionized water in order to give a pure product. $[C_5(MIM)_2][PF_6]_2$ was easily miscible with water; therefore, the loss of product in the aqueous layer occurred. In order to confirm this, the filtrate and washing liquid were collected and concentrated by rotary evaporation to obtain approximately 30% of $[C_5(MIM)_2][PF_6]_2$.

## 3.2. Melting points of dicationic ILs

In this study, three main factors were found to affect the melting points of dicationic ILs. They were (1) the dicationic nature (imidazolium or pyridinium), (2) the length of the linkage chains connecting the dications and (3) the symmetry of the dicationic ILs.

rsos.royalsocietypublishing.org    R. Soc. open sci. **5**: 181230

**Table 1.** Melting points and thermal stability of dicationic ionic liquids. $T_{onset}$: start of decomposition temperature. $W_{onset}$: remaining mass at $T_{onset}$.

| structure | Mp (°C) | $T_{onset}$ (°C) | $W_{onset}$ (%) |
|---|---|---|---|
| [C$_4$(MIM)$_2$][PF$_6$]$_2$ | 120–121 | 445 | 77 |
| [C$_5$(MIM)$_2$][PF$_6$]$_2$ | 85–86 | 444–445 | 73.5 |
| [C$_6$(MIM)$_2$][PF$_6$]$_2$ | 136 | 443–444 | 75 |
| [C$_4$(PYR)$_2$][PF$_6$]$_2$ | 276 | 414–415 | 72.5 |
| [C$_4$(MIM)][PF$_6$] | – | 426–427 | 81.5 |
| [C$_4$(PYR)][PF$_6$] | 79–80 | 395–396 | 78 |

$T_{onset}$: start of decomposition temperature

$W_{onset}$: remaining mass at $T_{onset}$

Considering first, dicationic ILs generate large molecules and high symmetry of dications, thus, they should have higher melting points than monocationic ILs. For example, [C$_4$(MIM)]PF$_6$ is liquid at room temperature, while [C$_4$(MIM)$_2$][PF$_6$]$_2$ is solid. Moreover, imidazolium-based ILs exhibited lower melting points than pyridinium-based ILs (table 1) for both monocationic ILs and dicationic ILs. An increase in the asymmetric structure of imidazolium-based salts improved the asymmetric disruption and ionic charge packing distortion, resulting in the reduction of melting points.

In addition to the effect of the different length of linkage chains, the longer linkage chain led to the higher melting points. As expected, the higher melting point was observed for [C$_6$(MIM)$_2$][PF$_6$]$_2$ that contained a longer alkyl chain than [C$_4$(MIM)$_2$][PF$_6$]$_2$. However, the melting point of [C$_5$(MIM)$_2$][PF$_6$]$_2$ was significantly lower than [C$_4$(MIM)$_2$][PF$_6$]$_2$ (table 1). A reduction in the symmetry of cations (five carbons in alkyl chain) caused a distortion from close-packing of the ionic charges in the solid-state lattice and caused the depression of melting points.

## 3.3. Thermal stability of dicationic ILs

The thermal stability of dicationic ILs was measured using TGA. TGA curves are presented in figure 1 and the results are summarized in table 1. As reported, the thermal decomposition temperature of monocationic ILs containing 1-alkyl-3-methylimidazolium cations with PF$_6^-$ as the anion ranged from 330 to 400°C [29]. In this study, [C$_4$(MIM)]PF$_6$ presented $T_{onset}$ at 426–427°C. Dicationic imidazolium-based ILs generated the thermal stability in a range of 443–445°C, which was higher than

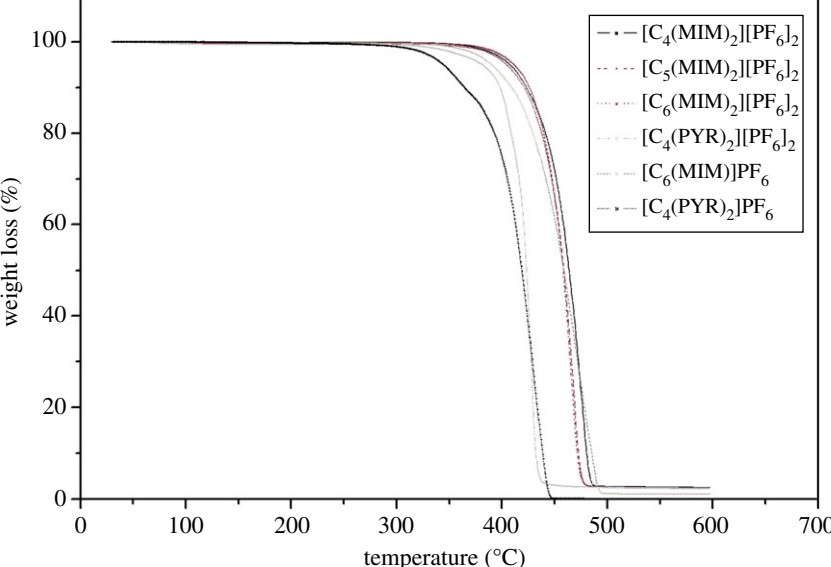

**Figure 1.** TGA curves of dicationic ionic liquids.

monocationic ILs. In a similar case, dicationic $[C_4(PYR)_2][PF_6]_2$ exhibited $T_{onset}$ at 414°C, which was higher than monocationic $[C_4(PYR)]$ $PF_6$ at 395–396°C. It was described that the thermal stability of 39 types of dicationic ILs was found to be significantly higher than those that have been observed for many traditional imidazolium-based ILs. The highest thermal stability (greater than 400°C was obtained with $[C_9(mpy)_2]NTf_2$ (1-methylpyrrolidinium dication) and the lowest decomposition temperature (330°C) was observed for $[C_9(bpy)_2]NTf_2$ (1-butylpyrrolidium dication) [14].

In addition, $[C_4(MIM)_2][PF_6]_2$ presented the higher $T_{onset}$ at 445°C and $[C_4(PYR)_2][PF_6]_2$ showed the lower $T_{onset}$ at 414°C. Imidazolium-based ILs had higher thermal stability than pyridinium-based ionic liquid, which was consistent with the previous study by Siedlecka *et al.* [30]. The thermal stability of ILs depends on the nucleophilicity of the cation and anion. Imidazolium-based ILs have higher nucleophilicity than pyridinium-based ILs and thus have higher thermal stability.

Furthermore, dicationic imidazolium-based ILs with different linkage chain generated the thermal stability in a temperature range of 443–445°C with $W_{onset}$ between 73% and 77%. No significant impact of the linkage chain of dications on the thermal stability was observed. High thermal stability of dicationic ILs will undoubtedly ensure their utilization for widely variable applications.

## 3.4. Solubility of dicationic ILs in organic solvents

The solubility of dicationic ILs with common organic solvents is summarized in table 2. All of the dicationic ILs exhibited good solubility in DMSO, which was due to the basic dipole force between these ionic compounds and the polar solvent [31]. None of the ILs was soluble in organic solvents including ethyl acetate, dichloromethane, methanol, ethanol, ether, toluene and petroleum ether.

Specifically, dicationic ILs exhibited different solubility properties with acetonitrile and acetone. For instance, 1 g of $[C_4(MIM)_2][PF_6]_2$ or $[C_5(MIM)_2][PF_6]_2$ was easily dissolved in less than 1 ml of acetonitrile, while 1–10 ml was necessary for $[C_6(MIM)_2][PF_6]_2$ and 16 ml was required for $[C_4(PYR)_2][PF_6]_2$. Additionally, 1 g of $[C_5(MIM)_2][PF_6]_2$ was soluble in 1 ml of acetone. However, 1 g of $[C_4(PYR)_2][PF_6]_2$ was not dissolved in 100 ml of acetone. Imidazolium-based ILs were very miscible with acetone and acetonitrile than pyridinium-based ILs.

## 3.5. A homogeneous and heterogeneous system of dicationic ILs with water

In general, the solubility behaviour of dicationic ILs with common solvents was quite similar to that of monocationic ILs. It was expected that both dicationic and monocationic ILs with $PF_6^-$ as the anion were immiscible with water. However, in this study, we found that dicationic ILs could form a homogeneous or heterogeneous system with water, depending on the dealing temperature.

**Table 2.** Solubility of dicationic ILs in common organic solvents. IL1: [C$_4$(MIM)$_2$][PF$_6$]$_2$, IL2: [C$_5$(MIM)$_2$][PF$_6$]$_2$, IL3: [C$_6$(MIM)$_2$][PF$_6$]$_2$, IL4: [C$_4$(PYR)$_2$][PF$_6$]$_2$. '+++': 1 g ionic liquid was easily dissolved in the solvent (less than 1 ml). '++': 1 g ionic liquid was easily dissolved in the solvent (1−10 ml). '+': 1 g ionic liquid was easily dissolved in the solvent (10−100 ml). '−': 1 g ionic liquid was not dissolved in the solvent (more than 100 ml).

| | EtoAc | CH$_3$CN | acetone | CH$_2$Cl$_2$ | MeOH | EtOH | Et$_2$O | DMSO | toluene | PET (60−90°C) |
|---|---|---|---|---|---|---|---|---|---|---|
| IL1 | − | +++ | ++ | − | − | − | − | +++ | − | − |
| IL2 | − | +++ | +++ | − | − | − | − | +++ | − | − |
| IL3 | − | ++ | ++ | − | − | − | − | ++ | − | − |
| IL4 | − | + | − | − | − | − | − | ++ | − | − |

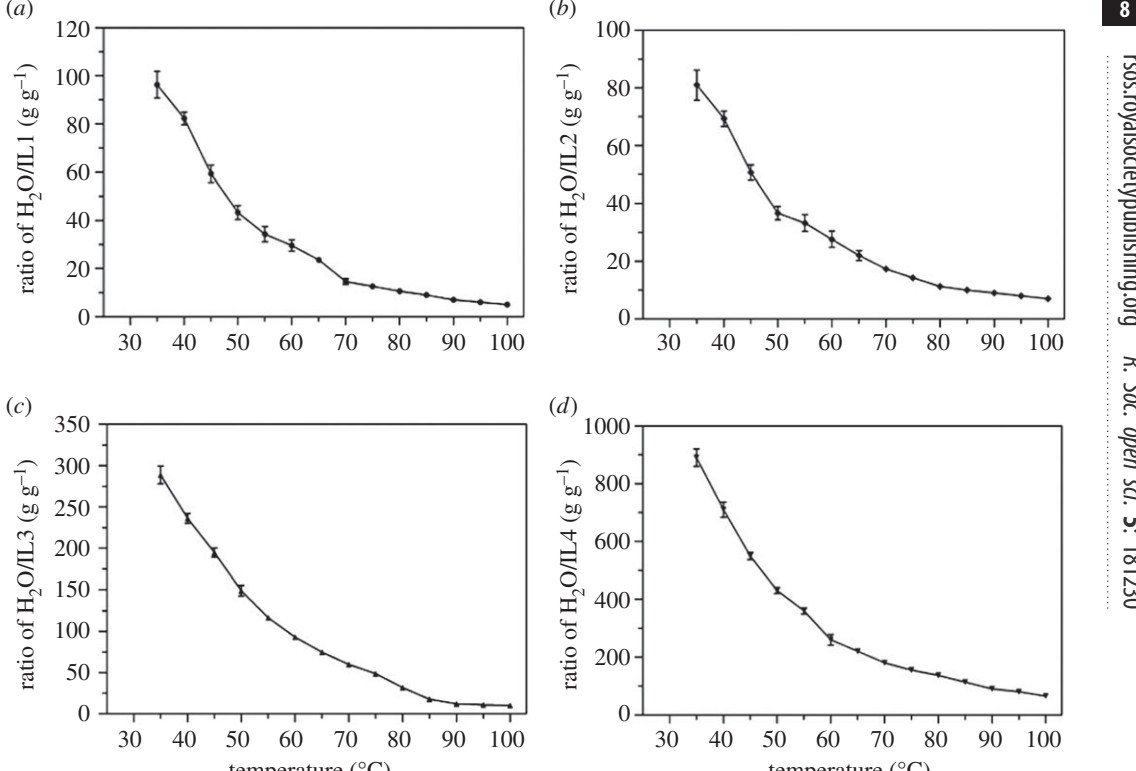

**Figure 2.** Miscibility of dicationic ILs and water at different temperatures. (*a*) [C$_4$(MIM)$_2$][PF$_6$]$_2$, (*b*) [C$_5$(MIM)$_2$][PF$_6$]$_2$, (*c*) [C$_6$(MIM)$_2$][PF$_6$]$_2$, (*d*) [C$_4$(PYR)$_2$][PF$_6$]$_2$.

As shown in figure 2, when the temperature was higher, the miscibility of dicationic ILs with water was better and the required amount of water was reduced. For example, 1 g of [C$_4$(MIM)$_2$][PF$_6$]$_2$ was miscible with 5 ml of water at 100°C, while 96 ml of water was required at room temperature. In a similar case, 7 ml of water was used for the miscibility of 1 g of [C$_5$(MIM)$_2$][PF$_6$]$_2$ at 100°C, while 80 ml of water was required at room temperature. Good miscibility was observed with imidazolium-based dicationic ILs (figure 2*a*–*c*) than pyridinium-based ILs (figure 2*d*).

As the linkage chain of dications was increased, the miscibility of dicationic ILs with water was reduced (figure 2). At 100°C, 5 ml of water was used to form a homogeneous system with 1 g of [C$_4$(MIM)$_2$][PF$_6$]$_2$. However, 10 ml was required for 1 g of [C$_6$(MIM)$_2$][PF$_6$]$_2$, as an increase in the linkage chain improved the hydrophobic character of [C$_6$(MIM)$_2$][PF$_6$]$_2$. Specifically, higher miscibility was observed with [C$_5$(MIM)$_2$][PF$_6$]$_2$ (figure 2*b*) than [C$_4$(MIM)$_2$][PF$_6$]$_2$ (figure 2*a*). It was possibly due to its lower symmetry of cations that generated good miscibility with water.

Interestingly, dicationic ILs could form a homogeneous system with a relatively low amount of water and acetonitrile at room temperature with the following ratios: [C$_4$(MIM)$_2$][PF$_6$]$_2$ : H$_2$O : CH$_3$CN (1 g : 1 ml : 3 ml), [C$_5$(MIM)$_2$][PF$_6$]$_2$ : H$_2$O : CH$_3$CN (1 g : 1 ml : 3 ml), [C$_6$(MIM)$_2$][PF$_6$]$_2$ : H$_2$O : CH$_3$CN (1 g : 1 ml : 4 ml). In such cases, imidazolium-based ILs were ready to form a homogeneous solution with water and acetonitrile at room temperature. When acetonitrile was removed from the mixture by rotary evaporation, dicationic ILs could be recovered from the solution readily by filtration at room temperature.

## 3.6. The recovery of dicationic ILs

The recovery of dicationic ILs is important for their applications in various research areas. In this work, the recovery of ILs by altering the temperature from ILs/H$_2$O solution at 100°C was determined. At 100°C, the clear solution of ILs with an appropriate amount of water was formed. When the temperature was cooled to room temperature, the ionic liquid was regenerated from the solution immediately and was filtered out simply.

As shown in figure 3, high percentage yields of [C$_4$(MIM)$_2$][PF$_6$]$_2$ (97.6%), [C$_5$(MIM)$_2$][PF$_6$]$_2$ (97.3%), [C$_6$(MIM)$_2$][PF$_6$]$_2$ (98.0%) and [C$_4$(PYR)$_2$][PF$_6$]$_2$ (94.2%) were achieved. It indicated that these dicationic

rsos.royalsocietypublishing.org    R. Soc. open sci. **5**: 181230

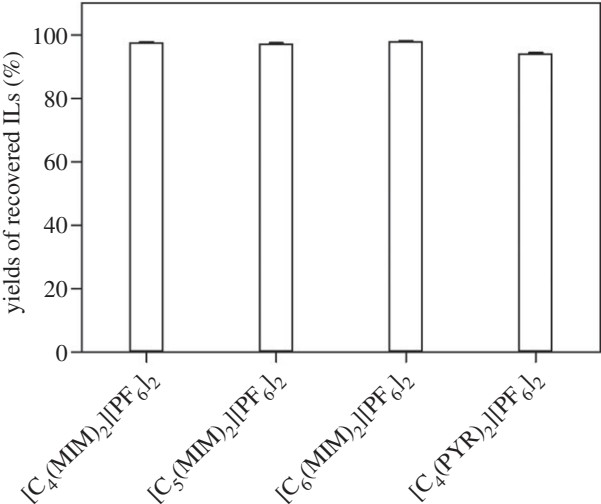

**Figure 3.** The recovery of dicationic ILs.

ILs were effective to form a homogeneous system with water at a relatively high temperature and then to form heterogeneous system when the solution was cooled to room temperature, which might benefit some particular reactions involving water. Since monocationic water-miscible ILs are difficult to recycle from aqueous solution, dicationic ILs with high recovery yield and high purity will provide new opportunities.

# 4. Conclusion

Dicationic ILs generate high thermal stability and they are suitable for utilizations and applications at high temperature conditions. Dicationic ILs have important implications in the design of homogeneous and heterogeneous systems with water and organic solvents, which will greatly benefit some specific applications in order to enable the effective separation and isolation of products. Dicationic ILs are recovered readily and efficiently, which have great potential for their industrial applications.

Ethics. All of our research was not concerning animal ethics.

Data accessibility. Our data are deposited at Dryad Digital Repository: https://dx.doi.org/10.5061/dryad.r1b2348 [32].

Authors' contributions. Y.-X.S. designed our experiments and completed most of the experiments such as the synthesis of ILs and the determination of water solubility and processed the experimental data. Y.-Y.W. and B.-B.S. measured the solubility of ILs in organic solvents and participated in the drawing process. B.-X.Z. put forward key opinions on the revision of the paper. X.-M.H., as the author of the correspondence, provided great help in the early design of the experiment and wrote and revised the article.

Competing interests. We have no competing interests.

Funding. This work was supported financially by National Science Foundation of China (21506030) and 'Young Talent' Project of Northeast Agricultural University in China (16QC30) and The 8th Special Financial Grant from the China Postdoctoral Science Foundation (2015T80375).

Acknowledgements. We are grateful to College of Life Sciences of Northeast Agricultural University for providing equipment supports.

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
