## [Reviewer comments · Royal Society Open Science]

Review History

RSOS-180708.R0 (Original submission)

Review form: Reviewer 1 (Maria Timofeeva)

Is the manuscript scientifically sound in its present form?

Yes

Are the interpretations and conclusions justified by the results?

Yes

Is the language acceptable?

Yes

Is it clear how to access all supporting data?

Yes

Do you have any ethical concerns with this paper?

No

Have you any concerns about statistical analyses in this paper?

No

Recommendation?

Major revision is needed (please make suggestions in comments)

Comments to the Author(s)

Dear Authors

In your report you demonstrated the results of investigation of physicochemical properties of Dicationic Ionic Liquids, such of [C4(MIM)2][PF6]2, [C5(MIM)2][PF6]2, [C6(MIM)2][PF6]2 and [C4(PYR)2][PF6]2. Results are useful and required for the chemists working with ILs. Unfortunately, several comments and questions appeared after reading manuscript.

Title

In my opinion Title of manuscript does not demonstrates the main goal of your investigation. I suggest to change Title, for example, Synthesis and investigation of physicochemical properties of dicationic ionic liquids.

Introduction

The main goal of investigation should be described in detail. At present the main problems in chemistry of studied IL are unclear

Results and discussion

- Page 3, Lines 20-21. You wrote that "As mentioned in the experimental section, yields of ionic liquids followed the order of [C6(MIM)2][PF6]2 (90.39%) > [C4(MIM)2][PF6]2 (73.66%) > [C5(MIM)2][PF6]2 (57.66%)". I am not sure that you can write 77.66 or 90.39%, because the low precision of measurement. Probably, it should be 90, 78 and 58%

-

- Figure2 (Miscibility of dicationic ILs and water at different temperatures). In this Figure you demonstrated correlations between amount content (ml) and temperature. It will be better to re-build Y-axis as ratio of H2O/ILs (g/ml or g/g)

- Figure3. Caption is unclear. As I understand you allocate IL from ILs/H2O solution at 100oC by cooling solution. Note that amount of H2O in mixture should be added

I suggest to revise, for example, The recovery of dicationic ILs

- Figure 4 (Miscibility of dicationic ILs with water and acetonitrile at room temperature). Figure is unclear. I think that this Figure can be deleted.

Review form: Reviewer 2

Is the manuscript scientifically sound in its present form?

Yes

Are the interpretations and conclusions justified by the results?

No

Is the language acceptable?

No

Is it clear how to access all supporting data?

Not Applicable

Do you have any ethical concerns with this paper?

No

Have you any concerns about statistical analyses in this paper?

No

Recommendation?

Reject

Comments to the Author(s)

In this manuscript, Sun, Y. et al. prepared a series of dicationic ionic liquids and investigated the miscibilities of these ionic liquids with water. They tested the thermal stability and melting points of these ionic liquids and studied their solubilities and recovery rates. In this re-submitted version, authors didn't address my previous comments seriously and properly. The writing is worse, and authors even didn't delete the journal template instruction in the second paragraph. Many spaces between words are missed. More importantly, in my previous comments, I asked the authors to explain why monocationic pyridinium-based ILs had higher decomposition temperature than that of dicationic pyridinium-based ILs with the same anion, while dicationic imidazolium-based ILs had higher decomposition temperatures than that of monocationic imidazolium-based ILs. It is not a suitable way to address this comment by simply deleting the discussion of the decomposition temperatures of monocationic and dicationic pyridinium-based ILs. Based on the above comments, I don't consider this manuscript as a serious scientific paper.

Decision letter (RSOS-180708.R0)

09-Jul-2018

Dear Mr Sun:

Manuscript ID: RSOS-180708

Title: "Homogeneous and Heterogeneous System : Dicationic Ionic Liquids with Water"

Thank you for submitting the above manuscript to Royal Society Open Science. Your paper was sent to reviewers and their comments are included at the bottom of this letter.

In view of the concerns raised by the reviewers, the manuscript has been rejected in its current form. However, a new manuscript may be submitted which takes into consideration these comments.

Please note that resubmitting your manuscript does not guarantee eventual acceptance, and that your resubmission will be subject to peer review before a decision is made.

Your resubmitted manuscript should be submitted by 06-Jan-2019. If you are unable to submit by this date please contact the Editorial Office.

Yours sincerely,
Dr Laura Smith, MRSC
Publishing Editor, Journals
Royal Society of Chemistry,
Thomas Graham House,
Science Park, Milton Road,
Cambridge, CB4 0WF, UK

Royal Society Open Science - Chemistry Editorial Office

On behalf of the Subject Editor Professor Anthony Stace and the Associate Editor Professor Eva Hevia

REVIEWER(S) REPORTS:

Associate Editor Comments to Author ():

RSC Associate Editor:

Comments to the Author:

(There are no comments.)

RSC Subject Editor:

Comments to the Author:

(There are no comments.)

Reviewers' Comments to Author:

Reviewer: 1

Comments to the Author(s)

Dear Authors

In your report you demonstrated the results of investigation of physicochemical properties of Dicationic Ionic Liquids, such of [C4(MIM)2][PF6]2, [C5(MIM)2][PF6]2, [C6(MIM)2][PF6]2 and [C4(PYR)2][PF6]2. Results are useful and required for the chemists working with ILs. Unfortunately, several comments and questions appeared after reading manuscript.

Title

In my opinion Title of manuscript does not demonstrates the main goal of your investigation. I suggest to change Title, for example, Synthesis and investigation of physicochemical properties of dicationic ionic liquids.

Introduction

The main goal of investigation should be described in detail. At present the main problems in chemistry of studied IL are unclear

Results and discussion

- Page 3, Lines 20-21. You wrote that "As mentioned in the experimental section, yields of ionic liquids followed the order of [C6(MIM)2][PF6]2 (90.39%) > [C4(MIM)2][PF6]2 (73.66%) > [C5(MIM)2][PF6]2 (57.66%)". I am not sure that you can write 77.66 or 90.39%, because the low precision of measurement. Probably, it should be 90, 78 and 58%

-

- Figure2 (Miscibility of dicationic ILs and water at different temperatures). In this Figure you demonstrated correlations between amount content (ml) and temperature. It will be better to re-build Y-axis as ratio of H2O/ILs (g/ml or g/g)

- Figure3. Caption is unclear. As I understand you allocate IL from ILs/H2O solution at 100oC by cooling solution. Note that amount of H2O in mixture should be added

I suggest to revise, for example, The recovery of dicationic ILs

- Figure 4 (Miscibility of dicationic ILs with water and acetonitrile at room temperature). Figure is unclear. I think that this Figure can be deleted.

Reviewer: 2

Comments to the Author(s)

In this manuscript, Sun, Y. et al. prepared a series of dicationic ionic liquids and investigated the miscibilities of these ionic liquids with water. They tested the thermal stability and melting points of these ionic liquids and studied their solubilities and recovery rates. In this re-submitted version, authors didn't address my previous comments seriously and properly. The writing is worse, and authors even didn't delete the journal template instruction in the second paragraph. Many spaces between words are missed. More importantly, in my previous comments, I asked the authors to explain why monocationic pyridinium-based ILs had higher decomposition temperature than that of dicationic pyridinium-based ILs with the same anion, while dicationic imidazolium-based ILs had higher decomposition temperatures than that of monocationic imidazolium-based ILs. It is not a suitable way to address this comment by simply deleting the discussion of the decomposition temperatures of monocationic and dicationic pyridinium-based ILs. Based on the above comments, I don't consider this manuscript as a serious scientific paper.

Author's Response to Decision Letter for (RSOS-180708.R0)

See Appendix A.

RSOS-181230.R0

Review form: Reviewer 1 (Maria Timofeeva)

Is the manuscript scientifically sound in its present form?

Yes

Are the interpretations and conclusions justified by the results?

Yes

Is the language acceptable?

Yes

Is it clear how to access all supporting data?

Yes

Do you have any ethical concerns with this paper?

No

Have you any concerns about statistical analyses in this paper?

No

Recommendation?

Accept with minor revision (please list in comments)

Comments to the Author(s)

Dear Authors

Unfortunately, several moments still need to be revised

Introduction

I suggest to specify the main goal of your investigation. For example, "The main goal of this investigation was to increase knowledge in physicochemical properties of dicationic Ionic Liquids. The main attentions were focused on the investigations (a) its thermal stability, (b) its melting points and (c) its solubility in water and water-organic mixtures. All these parameters can be important in a future their applying as solvents or catalysts".

Figure3 was not revised. What do you mean when write "IL recovery rate" (Y-axis)? Percent (%) can be dimension of rate (!!!).

Review form: Reviewer 2

Is the manuscript scientifically sound in its present form?

Yes

Are the interpretations and conclusions justified by the results?

Yes

Is the language acceptable?

Yes

Is it clear how to access all supporting data?

Not Applicable

Do you have any ethical concerns with this paper?

No

Have you any concerns about statistical analyses in this paper?

No

Recommendation?

Accept as is

Comments to the Author(s)

In this revised manuscript, authors have carefully revised their manuscript, corrected the format of the paper, and provided the detailed explanation to address my previous comments. So I recommend it can be accepted for publication.

Decision letter (RSOS-181230.R0)

13-Sep-2018

Dear Mr Sun:

Title: Synthesis and Investigation of Physicochemical Properties of Dicationic Ionic Liquids
Manuscript ID: RSOS-181230

Thank you for submitting the above manuscript to Royal Society Open Science. On behalf of the Editors and the Royal Society of Chemistry, I am pleased to inform you that your manuscript will be accepted for publication in Royal Society Open Science subject to minor revision in accordance with the referee suggestions. Please find the reviewers' comments at the end of this email.

The reviewers and handling editors have recommended publication, but also suggest some minor revisions to your manuscript. Therefore, I invite you to respond to the comments and revise your manuscript.

Please also include the following statements alongside the other end statements. As we cannot publish your manuscript without these end statements included, if you feel that a given heading is not relevant to your paper, please nevertheless include the heading and explicitly state that it is not relevant to your work. We have included a screenshot example of the end statements for reference.

- Ethics statement

Please clarify whether you received ethical approval from a local ethics committee to carry out your study. If so please include details of this, including the name of the committee that gave consent in a Research Ethics section after your main text. Please also clarify whether you received informed consent for the participants to participate in the study and state this in your Research Ethics section.

OR

Please clarify whether you obtained the necessary licences and approvals from your institutional animal ethics committee before conducting your research. Please provide details of these licences and approvals in an Animal Ethics section after your main text.

OR

Please clarify whether you obtained the appropriate permissions and licences to conduct the fieldwork detailed in your study. Please provide details of these in your methods section.

Because the schedule for publication is very tight, it is a condition of publication that you submit

the revised version of your manuscript before 22-Sep-2018. Please note that the revision deadline will expire at 00.00am on this date. If you do not think you will be able to meet this date please let me know immediately.

Best wishes,
Dr Laura Smith, MRSC
Publishing Editor, Journals
Royal Society of Chemistry,
Thomas Graham House,
Science Park, Milton Road,
Cambridge, CB4 0WF, UK
Tel: +44 (0)1223 438301

www.rsc.org

Royal Society Open Science - Chemistry Editorial Office

On behalf of the Subject Editor Professor Anthony Stace and the Associate Editor Professor Eva Hevia.

RSC Associate Editor
Comments to the Author:
(There are no comments.)

Reviewer comments to Author:
Reviewer: 1

Comments to the Author(s)
Dear Authors
Unfortunately, several moments still need to be revised

Introduction

I suggest to specify the main goal of your investigation. For example, "The main goal of this investigation was to increase knowledge in physicochemical properties of dicationic Ionic Liquids. The main attentions were focused on the investigations (a) its thermal stability, (b) its melting points and (c) its solubility in water and water-organic mixtures. All these parameters can be important in a future their applying as solvents or catalysts".

Figure3 was not revised. What do you mean when write "IL recovery rate" (Y-axis)? Percent (%) can be dimension of rate (!!!).

Reviewer: 2

Comments to the Author(s)
In this revised manuscript, authors have carefully revised their manuscript, corrected the format of the paper, and provided the detailed explanation to address my previous comments. So I recommend it can be accepted for publication.

Author's Response to Decision Letter for (RSOS-181230.R0)

See Appendix B.

RSOS-181230.R1 (Revision)

Review form: Reviewer 1 (Maria Timofeeva)

Is the manuscript scientifically sound in its present form?

Yes

Are the interpretations and conclusions justified by the results?

Yes

Is the language acceptable?

Yes

Is it clear how to access all supporting data?

Not Applicable

Do you have any ethical concerns with this paper?

No

Have you any concerns about statistical analyses in this paper?

I do not feel qualified to assess the statistics

Recommendation?

Accept with minor revision (please list in comments)

Comments to the Author(s)

Dear Authors

Unfortunately, several moments still need to be revised

- Page 1, Lines 29-32. You added phrase "The recovered ILs by altering the temperature were obtained in high yields of [C4(MIM)2][PF6]2 (0.98 g), [C5(MIM)2][PF6]2 (0.97 g), [C6(MIM)2][PF6]2 (0.98 g) and [C4(PYR)2][PF6]2 (0.94 g)." I suggest to write percentage yield.

- Page 4, Lines 42-44. You wrote "As shown in Figure 3, high yields of [C4(MIM)2][PF6]2 (0.98 g), [C5(MIM)2][PF6]2 (0.97 g), [C6(MIM)2][PF6]2 (0.98 g) and [C4(PYR)2][PF6]2 (0.94 g) were achieved." I suggest to write percentage yield

- Figure3 was revised; however, it will be better to write "Yield, (%)".

Review form: Reviewer 2

Is the manuscript scientifically sound in its present form?

Yes

Are the interpretations and conclusions justified by the results?

Yes

Is the language acceptable?

Yes

Is it clear how to access all supporting data?

Not Applicable

Do you have any ethical concerns with this paper?

No

Have you any concerns about statistical analyses in this paper?

No

Recommendation?

Accept as is

Comments to the Author(s)

In this revised manuscript, authors have carefully revised their manuscript according to previous comments. I recommend it can be accepted for publication..

Decision letter (RSOS-181230.R1)

09-Oct-2018

Dear Mr Sun:

Title: Synthesis and Investigation of Physicochemical Properties of Dicationic Ionic Liquids
Manuscript ID: RSOS-181230.R1

Thank you for submitting the above manuscript to Royal Society Open Science. On behalf of the Editors and the Royal Society of Chemistry, I am pleased to inform you that your manuscript will be accepted for publication in Royal Society Open Science subject to minor revision in accordance with the referee suggestions. Please find the reviewers' comments at the end of this email.

The reviewers and handling editors have recommended publication, but also suggest some minor revisions to your manuscript. Therefore, I invite you to respond to the comments and revise your manuscript.

Because the schedule for publication is very tight, it is a condition of publication that you submit the revised version of your manuscript before 18-Oct-2018. Please note that the revision deadline will expire at 00.00am on this date. If you do not think you will be able to meet this date please let me know immediately.

Best wishes,

Dr Laura Smith, MRSC
Publishing Editor, Journals
Royal Society of Chemistry,
Thomas Graham House,
Science Park, Milton Road,
Cambridge, CB4 0WF, UK

Royal Society Open Science - Chemistry Editorial Office

On behalf of the Subject Editor Professor Anthony Stace and the Associate Editor Professor Eva Hevia.

RSC Associate Editor:
Comments to the Author:
(There are no comments.)

RSC Subject Editor:
Comments to the Author:

(There are no comments.)

Reviewer comments to Author:

Reviewer: 2

Comments to the Author(s)

In this revised manuscript, authors have carefully revised their manuscript according to previous comments. I recommend it can be accepted for publication..

Reviewer: 1

Comments to the Author(s)

Dear Authors

Unfortunately, several moments still need to be revised

- Page 1, Lines 29-32. You added phrase "The recovered ILs by altering the temperature were obtained in high yields of [C4(MIM)2][PF6]2 (0.98 g), [C5(MIM)2][PF6]2 (0.97 g),

[C6(MIM)2][PF6]2 (0.98 g) and [C4(PYR)2][PF6]2 (0.94 g)." I suggest to write percentage yield.

- Page 4, Lines 42-44. You wrote "As shown in Figure 3, high yields of [C4(MIM)2][PF6]2 (0.98 g),

[C5(MIM)2][PF6]2 (0.97 g), [C6(MIM)2][PF6]2 (0.98 g) and [C4(PYR)2][PF6]2 (0.94 g) were

achieved." I suggest to write percentage yield

- Figure3 was revised; however, it will be better to write "Yield, (%)".

Author's Response to Decision Letter for (RSOS-181230.R1)

See Appendix C.

Decision letter (RSOS-181230.R2)

23-Oct-2018

Dear Mr Sun:

Title: Synthesis and Investigation of Physicochemical Properties of Dicationic Ionic Liquids

Manuscript ID: RSOS-181230.R2

It is a pleasure to accept your manuscript in its current form for publication in Royal Society Open Science. The chemistry content of Royal Society Open Science is published in collaboration with the Royal Society of Chemistry.

On behalf of the Subject Editor Professor Anthony Stace and the Associate Editor Professor Eva Hevia.

RSC Associate Editor
Comments to the Author:
Manuscript has now been revised accordingly to the reviewers comments.

Reviewer(s)' Comments to Author:

Appendix A

Dear Editors and Reviewers,

Thank you for your advice. All advice is important to us. We have corrected all the contents according to reviewers' suggestions. We also did further experiments. We really appreciated if you can give us an opportunity. Thank you for your help.

Responds to the reviewer's comments:

Reviewer 1:

Title

In my opinion Title of manuscript does not demonstrates the main goal of your investigation. I suggest to change Title, for example, Synthesis and investigation of physicochemical properties of dicationic ionic liquids.

Introduction

The main goal of investigation should be described in detail. At present the main problems in chemistry of studied IL are unclear.

Results and discussion

- Page 3, Lines 20-21. You wrote that "As mentioned in the experimental section, yields of ionic liquids followed the order of $[\text{C}_6(\text{MIM})_2][\text{PF}_6]_2$ (90.39%) $> [\text{C}_4(\text{MIM})_2][\text{PF}_6]_2$ (73.66%) $> [\text{C}_5(\text{MIM})_2][\text{PF}_6]_2$ (57.66%)". I am not sure that you can write 77.66 or 90.39%, because the low precision of measurement. Probably, it should be 90, 78 and 58%.

- Figure 2. (Miscibility of dicationic ILs and water at different temperatures). In this Figure you demonstrated correlations between amount content (ml) and temperature. It will be better to re-build Y-axis as ratio of $\text{H}_2\text{O}/\text{ILs}$ (g/ml or g/g).

- Figure 3. Caption is unclear. As I understand you allocate IL from $\text{ILs}/\text{H}_2\text{O}$ solution at 100 °C by cooling solution. Note that amount of H_2O in mixture should be added.

I suggest to revise, for example, The recovery of dicationic ILs.

- Figure 4. (Miscibility of dicationic ILs with water and acetonitrile at room temperature). Figure is unclear. I think that this Figure can be deleted.

1. Answer: we have changed the title as “Synthesis and Investigation of Physicochemical Properties of Dicationic Ionic Liquids”.
2. Answer: we re-write the introductions and make the reach goal clearer.
3. Answer: we have corrected the data as “90%, 78%, 58%”.
4. Answer: we have re-build the Y-axis of Figure 2 as the ratio of H₂O/ILs (g/g).
5. Answer: we have corrected the caption of Figure 3 as suggested.
6. Answer: Figure 4 was deleted as suggested.

Reviewer 2:

In this manuscript, Sun, Y. et al. prepared a series of dicationic ionic liquids and investigated the miscibilities of these ionic liquids with water. They tested the thermal stability and melting points of these ionic liquids and studied their solubilities and recovery rates. In this re-submitted version, authors didn't address my previous comments seriously and properly. The writing is worse, and authors even didn't delete the journal template instruction in the second paragraph. Many spaces between words are missed. More importantly, in my previous comments, I asked the authors to explain why monocationic pyridinium-based ILs had higher decomposition temperature than that of dicationic pyridinium-based ILs with the same anion, while dicationic imidazolium-based ILs had higher decomposition temperatures than that of monocationic imidazolium-based ILs. It is not a suitable way to address this comment by simply deleting the discussion of the decomposition temperatures of monocationic and dicationic pyridinium-based ILs. Based on the above comments, I don't consider this manuscript as a serious scientific paper.

Answer:

We are really sorry about the version of the manuscript. When we uploaded our manuscript, all the formations of the file were changed, because we used a “WPS” word system, leading to the missing of many spaces between words. It is our fault and we are sorry for the inconvenience to read. We really appreciate you still could give us an opportunity.

Thank you for your previous comments. You asked us to explain why monocationic pyridinium-based ILs had higher decomposition temperature than that of dicationic pyridinium-based ILs with the same anion, while dicationic imidazolium-based ILs had higher decomposition temperatures than that of monocationic imidazolium-based ILs. It is really not a suitable way to simply

deleting the discussion. However, could you give us an opportunity to explain it please?

In our original manuscript, we described the above discussion, because we cited a literature (Mahrova M, Pagano F, Pejakovic V, Valea A, Kalin M, Igartua A, Tojoet E. 2015 Pyridinium based dicationic ionic liquids as base lubricants or lubricant additives. *Tribol. Int.* **82**,245–54. doi:10.1016/j.triboint.2014.10.018). In this paper, it was described that “dicationic ILs were found to be less stable than those monocationic ILs.” For example, “monocationic [C₁Py][C₁SO₄] present the T_{d,onset} at 601 K, the DIL (dicationic) [C₁Py(CH₂OCH₂)₃C₁Py][C₁SO₄]₂ shows the T_{d,onset} at 483K. Monocationic [C₁Py][NTf₂] presented the T_{d,onset} at 704 K, the DIL (dicationic) [C₁Py(CH₂OCH₂)₃C₁Py][NTf₂] shows the T_{d,onset} at 483 K.” However, the author emphasized “Unexpected results were observed” with these two ILs, because very few pyridinium-based ionic liquids were found in this case compared with most other literatures. These dicationic pyridinium-based ILs have low thermal stability, because there is an ether bond in the linkage chain of their dications that is unstable, which are different with the chemical structure of ILs in our work. Thus, we cited wrong information in our original manuscript and that is why we deleted the discussion in the re-submitted manuscript. Fortunately, you help us to find our mistake and thank you very much for your reminding.

In order to identify the thermal stability of monocationic ILs and dicationic ILs for both imidazolium based ILs and pyridinium based ILs, we did further experiments to confirm this. In our work, [C₄(MIM)]PF₆ presented T_{onset} at 426–427 °C. Dicationic imidazolium-based ILs generated the thermal stability in a range of 443–445 °C, which were higher than monocationic ILs. In a similar case, dicationic [C₄(PYR)₂][PF₆]₂ exhibited T_{onset} at 414 °C, which was higher than monocationic [C₄(PYR)]PF₆ at 395–396 °C. It was reported that the thermal stability of 39 types of dicationic ionic liquids were found to be significantly higher than those that have been observed for many traditional imidazolium-based ionic liquids. (Anderson JL, Ding R, Ellern A, Armstrong DW. 2005 Structure and properties of high stability geminal dicationic ionic liquids. *J. Am. Chem. Soc.* **127**,593–604. doi:10.1021/ja046521u). In their paper, the highest thermal stability (> 400 °C) was obtained with [C₉(mpy)₂]NTf₂ (1-methylpyrrolidinium dication) and the lowest decomposition temperature (330 °C) was observed for [C₉(bpy)₂]NTf₂ (1-butylpyrrolidinium dication). Our work was consistent with Dr. Anderson et al. We have described all data in the manuscript. Thank you.

Your advice is very important to us. We really appreciate for all your help.

Yours sincerely,

Yi-Xin Sun

Corresponding author: Xiaomei Hu

Appendix B

Dear Sir or Madam,

Thank you for your letter and for the reviewers comments concerning our manuscript entitled “Synthesis and Investigation of Physicochemical Properties of Dicationic Ionic Liquids” (ID: RSOS-181230). Those comments are all valuable and very helpful for revising and improving our paper, as well as the important guiding significance to our researches. We have studied comments carefully and have made correction. Revised portion are marked in red in the paper. The main corrections in the paper and the responds to the reviewer’s comments are as following:

Responds to the reviewer’s comments:

Referee: 1

Introduction

I suggest to specify the main goal of your investigation. For example, “The main goal of this investigation was to increase knowledge in physicochemical properties of dicationic Ionic Liquids. The main attentions were focused on the investigations (a) its thermal stability, (b) its melting points and (c) its solubility in water and water-organic mixtures. All these parameters can be important in a future their applying as solvents or catalysts”.

We have corrected the Introduction according to reviewer’s advices.

Thank you for your help.

Figure3 was not revised. What do you mean when write “IL recovery rate” (Y-axis)? Percent (%) can be dimension of rate (!!).

We have modified Figure 3 and re-write the contents in 4.6.

Referee: 2

In this revised manuscript, authors have carefully revised their manuscript, corrected the format of the paper, and provided the detailed explanation to address my previous comments. So I recommend it can be accepted for publication.

Thanks very much for your kind work and consideration on publication of our paper. On behalf of my co-authors, we would like to express our great appreciation to reviewers.

Thank you and best regards.

Yours sincerely,

Yi-Xin Sun

Corresponding author: Xiaomei Hu

E-mail: 1031857030@qq.com

Appendix C

Dear Sir or Madam,

Thank you for your letter and for the reviewers comments concerning our manuscript entitled “Synthesis and Investigation of Physicochemical Properties of Dicationic Ionic Liquids” (ID: RSOS-181230.R1). Those comments are all valuable and very helpful for revising and improving our paper, as well as the important guiding significance to our researches. We have studied comments carefully and have made correction. Revised portion are marked in red in the paper. The main corrections in the paper and the responds to the reviewer’s comments are as following:

Responds to the reviewer’s comments:

Referee: 2

In this revised manuscript, authors have carefully revised their manuscript according to previous comments. I recommend it can be accepted for publication.

Answer: Thanks very much for your kind work and consideration on publication of our paper. On behalf of my co-authors, we would like to express our great appreciation to reviewers.

Thank you and best regards.

Referee: 1

- Page 1, Lines 29-32. You added phrase “The recovered ILs by altering the temperature were obtained in high yields of [C₄(MIM)₂][PF₆]₂ (0.98 g), [C₅(MIM)₂][PF₆]₂ (0.97 g), [C₆(MIM)₂][PF₆]₂ (0.98 g) and [C₄(PYR)₂][PF₆]₂ (0.94 g).” I suggest to write percentage yield.

- Page 4, Lines 42-44. You wrote “As shown in Figure 3, high yields of [C₄(MIM)₂][PF₆]₂ (0.98 g), [C₅(MIM)₂][PF₆]₂ (0.97 g), [C₆(MIM)₂][PF₆]₂ (0.98 g) and [C₄(PYR)₂][PF₆]₂ (0.94 g) were achieved.” I suggest to write percentage yield

Answer: We have corrected the Page 1 and Page 4 according to reviewer’s advices. Page 1 “The recovered ILs by altering the temperature were obtained in high percentage yields of [C₄(MIM)₂][PF₆]₂ (97.6%), [C₅(MIM)₂][PF₆]₂ (97.3%), [C₆(MIM)₂][PF₆]₂ (98.0%) and [C₄(PYR)₂][PF₆]₂ (94.2%).” Page 4 “As shown in Figure 3, high percentage yields of [C₄(MIM)₂][PF₆]₂ (97.6%), [C₅(MIM)₂][PF₆]₂ (97.3%), [C₆(MIM)₂][PF₆]₂ (98.0%) and [C₄(PYR)₂][PF₆]₂ (94.2%) were achieved.”

- Figure3 was revised; however, it will be better to write “Yield, (%)”.

Answer: We have modified Figure 3 and Y-axis is revised to “Yields of recovered ILs (%)”

If we still have mistake to understand the coments at this time, could you help us to tell us in more detail please, and we will try our best to correct it. Thank you for all your help. Thank you for your help.

Yours sincerely,

Yi-Xin Sun

E-mail: 1031857030@qq.com